# GLOBAL ADVERSARIAL ROBUSTNESS GUARANTEES FOR NEURAL NETWORKS

## ABSTRACT

We investigate global adversarial robustness guarantees for machine learning models. Specifically, given a trained model we consider the problem of computing the probability that its prediction at any point sampled from the (unknown) input distribution is susceptible to adversarial attacks. Assuming continuity of the model, we prove measurability for a selection of local robustness properties used in the literature. We then show how concentration inequalities can be employed to compute global robustness with estimation error upper-bounded by $\epsilon$, for any $\epsilon > 0$ selected a priori. We utilise the methods to provide statistically sound analysis of the robustness/accuracy trade-off for a variety of neural networks architectures and training methods on MNIST, Fashion-MNIST and CIFAR. We empirically observe that robustness and accuracy tend to be negatively correlated for networks trained via stochastic gradient descent and with iterative pruning techniques, while a positive trend is observed between them in Bayesian settings.

## 1 INTRODUCTION

Adversarial attacks are potentially imperceptible input manipulations that when applied to a test point result in misclassification. Since even state-of-the-art deep learning models have been shown susceptible to such attacks (Goodfellow et al., 2014), adversarial examples have raised serious concerns about the security and robustness of models learned from data (Biggio & Roli, 2018). As standard accuracy measures fail to capture model behaviour in an adversarial setting, the development of techniques capable of quantifying the robustness of machine learning models is an essential pre-condition for their application in safety-critical scenarios, where model failures have already led to fatal accidents (Yadron & Tynan, 2016). In such settings, we require models that are not only accurate, but also have guaranteed robust behaviour (Michelmore et al., 2019). However, while techniques for the computation of *local* adversarial robustness guarantees (i.e., specific to a particular test point) have been developed for a variety of machine learning models (Katz et al., 2017; Biggio & Roli, 2018; Cardelli et al., 2019b), to the best of our knowledge relatively few studies of *global* robustness (Bastani et al., 2016; Ruan et al., 2019; Webb et al., 2018) have been considered, and none rigorously quantify the probability that a neural network is robust to adversarial perturbations.

Given a neural network (NN) $f$ we investigate the *global* robustness properties of $f$. We start from a local notion of robustness specific to a point (e.g. robustness against bounded norm perturbations or adversarial attacks) and pose the problem of computing the probability that the prediction of $f$ in a test point $x$, sampled from a (possibly unknown) data distribution $P$, is robust according to a given local property. Unfortunately, exact computation of such global robustness measures is intractable. Nevertheless, we show how statistically sound approximations with quantifiable and arbitrarily small error can be computed by making use of concentration inequalities (Massart, 2007). Crucially, this allows us to provide *statistical guarantees* over the estimation of the global robustness measures, while keeping computations to a minimum. In order to do so, we prove that the global adversarial robustness property defined here is *measurable* for a range of commonly used notions of adversarial examples and NN architectures, and show how global robustness can be reformulated as a sum of independent and identically distributed random variables. Hence, bounds such as additive Chernoff or Hoeffding's inequality can be used to quantify the error with respect to an empirical estimator.

We utilise our techniques to statistically quantify the robustness of a variety of different neural network architectures and training methods, and experimentally study their robustness profile on

multiple datasets, including MNIST, Fashion-MNIST and CIFAR. In particular, we analyse the effects of iterative magnitude pruning on network robustness. Interestingly, while weight pruning may increase test set accuracy (as observed by Frankle & Carbin (2018)), we find that it does not have any positive effect on global robustness for the examples we analysed. We further empirically evaluate the robustness accuracy trade-off (Tsipras et al., 2018) for a wide selection of different network architectures and hyper-parameters, which allows us to compare different training regimes. We find that accuracy and robustness tend to be at odds in networks trained via stochastic gradient descent, whereas there seems to exist a positive correlation between accuracy and robustness in Bayesian settings. More specifically, for the datasets analysed, we observe that optimising a Bayesian network architecture for accuracy leads to networks that are more robust against adversarial examples.

In summary, this paper makes the following main contributions:

- We consider probabilistic global adversarial robustness measures for neural networks and prove their measurability.

- We show how concentration inequalities can be used to provide statistical guarantees on global robustness estimations with an *a priori* error bound.

- We investigate the trade off between robustness and accuracy for a variety of networks architectures and training methods. We provide empirical results that suggest that networks trained by Bayesian methods might be more robust than their deterministic counter-parts.

**Related Works.** Various methods, both heuristic and based on formal verification, have been derived to evaluate the local adversarial robustness of neural networks (Goodfellow et al., 2014; Wicker et al., 2018; Tramèr et al., 2017; Katz et al., 2017; Cardelli et al., 2019a; Carlini et al., 2017). In addition to local robustness, which focuses on a specific test point, global measures of robustness have also been discussed in the literature (Bastani et al., 2016; Ruan et al., 2019; Webb et al., 2018). Intuitively, these generalise the notion of local robustness to a set of test points (or a data distribution), so as to marginalise out their influence. This yields a global measure of robustness for the network that describes the probability wrt the input distribution that a certain local property is satisfied (Dreossi et al., 2019). While these approaches consider similar global robustness measures to those discussed here, they lack a characterisation of error bounds. In contrast, our approach provides statistical error bounds on the estimation of global robustness, up to any *a priori* specified tolerance $\epsilon > 0$.

Formal bounds on global adversarial robustness are given by (Fawzi et al., 2018). However, implementation details are given only for linear and quadratic classifiers, since the computation of the bound relies on the distance from the decision boundary. Our approach instead generalises to any continuous model for which local robustness can be computed. A worst-case notion of global robustness is discussed by Katz et al. (2017); Gopinath et al. (2018), where a network is considered *globally adversarially robust* if there are no input points that are vulnerable to adversarial attacks. However, this notion of global robustness tends to be too pessimistic and hard to compute in practice, since it is valid for all points in the input space (independently of their likelihood wrt the input distribution). Instead, the global robustness measure considered here is probabilistic, in that it computes the probability that a point drawn from the data distribution is vulnerable to adversarial attacks.

Similar techniques to those used in this paper are applied for the computation of PAC bounds for the evaluation of the generalisation error of learning models (McAllester, 1999; Vapnik, 2013) and their robustness (Xu & Mannor, 2012; Gourdeau et al., 2019). This differs from the problem studied here as PAC aims at bounding the generalisation capabilities of a family of learning models, whereas our method yields bounds on a specific trained model. Additionally, when computing global robustness, we focus on analyzing the probability that a perturbation applied to a test point causes a prediction change, independently of the point ground truth (i.e. independently of the generalisation error).

## 2 GLOBAL ADVERSARIAL ROBUSTNESS OF NEURAL NETWORKS

In this section we discuss two notions of global robustness for neural networks. Namely, we first introduce qualitative and quantitative variants of local robustness, and then define global robustness as their expected values wrt the (possibly unknown) input distribution. Throughout this paper, we consider a neural network $f : \mathbb{R}^m \to \mathbb{R}^n$, with any activation function and arbitrarily many layers, such that, for a test point $x \in \mathbb{R}^m$, $f(x) = (f_1(x), ..., f_n(x))$ represents the vector of the confidence

values for each of the $n$ class labels. $\mathcal{D} = \{(x,y)|x \in \mathbb{R}^m, y \in \{1, ..., n\}\}$ is a test dataset comprising $|\mathcal{D}|$ input points, which we assume to be iid sampled from a distribution $P$.

**A Measure of Global Robustness.** Given a test point $x$ sampled from $P$, we consider the indicator function $\mathrm{g} : \mathbb{R}^m \to \{0, 1\}$, which returns 1 if $f$ is *locally robust* in $x$, and is defined as follows.

**Definition 1.** *(Local Robustness) Given $x \in \mathbb{R}^m$ and $\delta > 0$, let $T^x = \{\bar{x} \in \mathbb{R}^m \,|\, |\bar{x} - x|_p \leq \delta\}$, where $|\cdot|_p$ is an $L_p$ norm, be a $\delta-$ball around $x$. Call $C(x) = \arg\max_{i \in \{1,...,n\}} f_i(x)$ the set of classes for which the confidence of $f$ in $x$ is maximised. Then we say that $f$ is locally robust in $x$ around $T^x$ iff $\mathrm{g}(x) = 1$, where:*

$$\mathrm{g}(x) = \begin{cases} 1 & \text{if } C(x) \text{ is a singleton} \quad \text{and} \quad \forall \bar{x} \in T^x \, C(\bar{x}) = C(x) \\ 0 & \text{otherwise.} \end{cases}$$

That is, $f$ is locally robust in $x$ if small perturbations in the input do not cause a classification change. Note that we require the predicted class of $x$ to be unambiguous, i.e. $C(x)$ comprises only one class, and otherwise we view the point as not locally robust (as the classification decision output is not uniquely defined in this case). Local robustness is widely used to investigate the properties of neural networks and various algorithms have been derived for its computation and approximation (Dreossi et al., 2019). Details on how we compute or approximate $\mathrm{g}(x)$ are given in Section 3.1.

Building on local robustness, we are interested in computing the *global robustness* of $f$, which is defined below as the probability that a test point sampled from $P$ is locally robust.

**Definition 2.** *(Global Robustness) The robustness of $f$ is defined as:*

$$R(\mathrm{g}) = \mathrm{E}_{x \sim P}[\mathrm{g}(x)] = \int \mathrm{g}(x)p(x)dx, \tag{1}$$

*where $p$ is the density probability associated to $P$.*

It is important to emphasise that $R(\mathrm{g})$ is a property of the neural network, as the effect of each single input point is marginalised out when taking the expectation. We remark that, since $\mathrm{g}$ is an indicator function, under the assumption that $\mathrm{g}$ is measurable (discussed in detail in Section 3), it follows that $R(\mathrm{g})$ is a well defined probability measure.

**A Quantitative Measure of Global Robustness.** As discussed in the previous section, we define local robustness $\mathrm{g}$ as an indicator function. It is often useful to consider a quantitative notion for the robustness of $f$. For example, given a $\delta-$ball around a test point $x$, we may want to know not just if $f$ is locally robust in $x$, but also the maximal variation in the classification confidence values of the various classes, which evaluate the network's robustness independently of the decision procedure used for the classification. Further, one could require the model confidence to be robust up to a specific output threshold (Katz et al., 2017). Specifically, given $f$ and a test point $x \in \mathbb{R}^m$, we consider a function $\bar{\mathrm{g}}(x)$ that quantifies the robustness of $f$ in $x$ and is defined as follows.

**Definition 3.** *(Quantitative Local Robustness) Given $x \in \mathbb{R}^m$ and $\delta > 0$, let $T^x = \{x \in \mathbb{R}^m \,|\, |\bar{x} - x|_p \leq \delta\}$, where $|\cdot|_p$ is an $L_p$ norm, be a $\delta-$ball around $x$. We define the quantitative local robustness of $f$ in $x$ around $T^x$ as:*

$$\bar{\mathrm{g}}(x) = \max_{\bar{x} \in T^x} h(f(x), f(\bar{x})),$$

*where $h : \mathbb{R}^n \times \mathbb{R}^n \to \mathbb{R}$ is a given function that measure differences between $f$ predictions.*

In Section 4 we provide experimental results using $h$ defined as the $L_\infty$ norm between predictions, as well as by checking whether this norm is greater than a given threshold $\gamma > 0$ (which is akin to the definition introduced by Katz et al. (2017)). Similarly to $R(\mathrm{g})$, the following definition generalises quantitative local robustness to a global property, by taking the expectation of $\bar{\mathrm{g}}(x)$ with respect to the input data distribution.

**Definition 4.** *(Quantitative Global Robustness) The quantitative robustness of $f$ is defined as:*

$$D(\bar{\mathrm{g}}) = \mathrm{E}_{x \sim P}[\bar{\mathrm{g}}(x)] = \int \bar{\mathrm{g}}(x)p(x)dx, \tag{2}$$

*where $p$ is the density probability associated to $P$.*

Exact computation of $R(\mathrm{g})$ and $D(\bar{\mathrm{g}})$ is infeasible as it requires the computation of an integral with respect to an unknown input distribution. Nevertheless, in what follows, we show that under mild assumptions it is possible to estimate these quantities with *a priori* arbitrarily stringent guarantees.

## 3    STATISTICAL GUARANTEES ON ADVERSARIAL ROBUSTNESS

In this section we derive estimations of $R(\mathrm{g})$ and $D(\bar{\mathrm{g}})$ with arbitrarily stringent *a priori* statistical guarantees. In particular, in Equation 3 we consider empirical estimators of these quantities and then in Theorem 1 and 2 we show that the probability that the error between these quantities and the real measures is greater than a threshold can be upper bounded by employing concentration inequalities. In Section 3.1 we review how g and $\bar{\mathrm{g}}$ are computed or approximated in practice, and finally describe the global robustness guarantees computation pipeline in Section 3.2. Proofs for the Propositions and Theorems state in this Section are reported in the Appendix Section A.

We consider the following empirical estimators:

$$R^{emp}(\mathrm{g}, S) = \sum_{(x,y)\in S} \frac{\mathrm{g}(x)}{|S|} \quad D^{emp}(\bar{\mathrm{g}}, S) = \sum_{(x,y)\in S} \frac{\bar{\mathrm{g}}(x)}{|S|} \tag{3}$$

where $S$ is a set of size $|S|$ of test points iid sampled from $P$. In order to derive a worst-case scenario bound on the distance between $R(\mathrm{g})$ and $R^{emp}(\mathrm{g}, S)$, we first have to show that g is a measurable function[1]. In fact, the measurability of g guarantees that $R(\mathrm{g})$ is a well defined probability measure induced by $P$, the data distribution. Hence, concentration inequalities can be applied to bound the error between $R(\mathrm{g})$ and $R^{emp}(\mathrm{g}, S)$.

**Proposition 1.** *Assume that, for $i \in \{1, ..., n\}$, $f_i$, the i-th component of the neural network $f$ is a continuous function, then $\mathrm{g}(x)$, as defined in Definition 1, is measurable.*

Proof of Proposition is based on noticing that the set $\{x \in \mathbb{R}^m \,|\, \mathrm{g}(x) > 0\}$ can be rewritten as the union of pre-images of measurable sets for a measurable function. Note that the overwhelming majority of the neural networks commonly used in practice are continuous. Hence, the assumption in Proposition 1 is almost always verified in practice. At this point, we can state the following theorem, which bounds the probability that the distance between $R(\mathrm{g})$ and $R^{emp}(\mathrm{g}, S)$ is greater than $\epsilon$, for any $\epsilon > 0$.

**Theorem 1.** *Assume that g is measurable, then for any $\epsilon > 0$ it holds that that for any (possibly unknown) input data distribution $P$*

$$Prob(|R^{emp}(\mathrm{g}, S) - R(\mathrm{g})| > \epsilon) \le 2e^{-2\epsilon^2|S|}. \tag{4}$$

Theorem 1 gives a bound that is *problem and architecture independent*, that is it holds for any data distribution and any architecture of the network $f$. As illustrated in Section 4.1, the required number of samples for a given error tolerance $\epsilon$, although exponential in $\epsilon$, is generally under control in practice. The proof of Theorem 1 relies on reformulating Equation 1 as a probability measure and then the application of the Chernoff bound to a sum of independent Bernoulli random variables.

In the remaining part of this section, we show how to derive statistical guarantees on the error between $D(\bar{\mathrm{g}})$ and $D^{emp}(\bar{\mathrm{g}}, S)$. Again, we first show that under mild assumptions $\bar{\mathrm{g}}(x)$ is measurable. Then, we use this result to bound the error by using concentration inequalities.

**Proposition 2.** *Consider $\bar{\mathrm{g}}$, as defined in Definition 3. Then, under the assumption that $l(x_1, x_2) = h(f(x_1), f(x_2))$ is continuous in both $x_1$ and $x_2$, we have that $\bar{\mathrm{g}}$ measurable.*

In the following theorem we show that a result similar to that of Theorem 1 can be obtained also for the more general case of quantitative global robustness.

**Theorem 2.** *Assume that $\bar{\mathrm{g}}$ is measurable and that, for any input point $x$, $\bar{\mathrm{g}}(x) \in [A, B] \subset \mathbb{R}^n$. Then, for any (possibly unknown) input data distribution $P$, it holds that*

$$Prob(|D^{emp}(\bar{\mathrm{g}}, S) - D(\bar{\mathrm{g}})| > \epsilon) \le 2e^{-\frac{2\epsilon^2|S|}{(B-A)^2}}. \tag{5}$$

Note that in practice it is often the case that the output of a neural network is bounded (notably for classification problems and regression problems over a bounded input space). The proof of Theorem 2 relies on the application of union bound and Hoeffding inequality (Vapnik, 2013), and the resulting upper bound is again problem and architecture independent.

---

[1]In this paper by measurable we mean Borel measurable.

## 3.1 Computation of Local Robustness $\mathbf{g}(x)$ and $\bar{\mathbf{g}}(x)$

The computation of global robustness depends on the ability to compute local robustness $\mathbf{g}(x)$ (or analogously $\bar{\mathbf{g}}(x)$) for every input point $x$, that is, to establish the presence or absence of an adversarial attack in a neighbourhood of $x$. For deep neural networks this is known to be NP-complete (Katz et al., 2017). For exact computation of $\mathbf{g}(x)$ we employ the verification method introduced by (Huang et al., 2017), which builds on input space discretisation and constraint solving to perform exhaustive search of the neighbourhood. Results for this are discussed in Section 4.1. Unfortunately, the computational complexity for the exact computation of $\mathbf{g}(x)$ quickly gets prohibitive for large networks. As in (Bastani et al., 2016), in these cases we proceed by approximating local robustness. The trade-off between scalability and approximation quality of an adversarial attack method is empirically discussed in (Carlini et al., 2017). We remark that the statistical bounds we compute are fully transparent to the way in which $\mathbf{g}$ is computed or approximated. More precisely, the methods discussed in Section 3 provide sound, statistical bounds on the global robustness estimation, independently on the definition and computation of $\mathbf{g}(x)$.

## 3.2 Computation of Global Robustness $R(\mathbf{g})$ and $D(\bar{\mathbf{g}})$

We detail how global robustness with statistical guarantees can be computed. In Theorems 1 and 2 we proved that statistical guarantees can be computed on standard estimators for $R(\mathbf{g})$ and $D(\bar{\mathbf{g}})$. In fact, given a statistical tolerance $\epsilon > 0$, this can be done by computing the smallest number of samples $N = |S|$ that satisfies Equation 4 (respectively Equation 5 for the computation of $D(\bar{\mathbf{g}})$). Let $S = \{x_i,\ i = 1, \dots, N\}$ be $N$ test points randomly taken from the test dataset $\mathcal{D}$, we then compute the values of $\mathbf{g}(x_i)$ (respectively $\bar{\mathbf{g}}(x_i)$) using the methods discussed in the previous subsection. We finally use these values to evaluate Equation 3. This provides us with an estimator $R^{emp}(\mathbf{g}, S)$ (respectively $D^{emp}(\bar{\mathbf{g}}, S)$) that meets the required statistical guarantees by construction.

## 4 Experiments and Applications

We apply the presented techniques to obtain statistically sound estimation of the robustness profile for an array of NN architectures. In Section 4.1 we analyse the convergence rate of the bound on global robustness, using the exact local robustness method of Huang et al. (2017). In Section 4.2 we report empirical robustness/accuracy trade-off for a variety of different networks trained on MNIST and CIFAR, using FGSM (Goodfellow et al., 2014) for approximate computation of local robustness. Section 4.3 analyses the effects of pruning on the network generalisation and global robustness on MNIST and Fashion-MNIST. Results for Bayesian networks are discussed in Section 4.4[2].

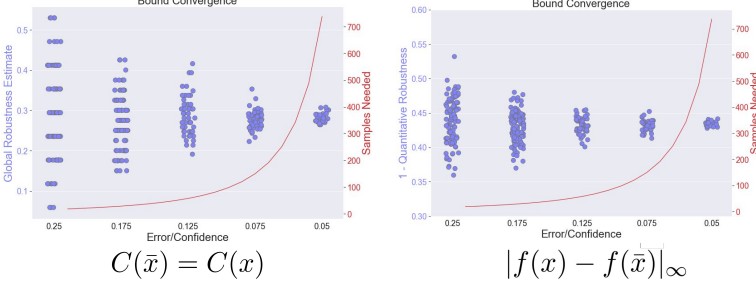

Figure 1: Convergence of the global robustness estimation wrt error tolerance $\epsilon$ (decreasing values along the x-axis). For each $\epsilon$ we compute 50 empirical estimates (blue dots). As the error tolerance decreases the estimated quantities converge to a tight cluster. The red line interpolates the number of sampled images needed for the computation of each blue dot.

## 4.1 Convergence Analysis of Bound

Figure 1 depicts the convergence rate of our robustness estimation method wrt the error tolerance $\epsilon$. Namely, we train a CNN on MNIST and estimate its global robustness up to a decreasing value of $\epsilon$.

---

[2]Code can be found at: *link left blank during peer review.*

For each value of $\epsilon$ we compute 50 different estimates of global robustness (blue dots in the figure), statistically guaranteed to be $\epsilon$-close to the actual robustness of the network. In order to do so, for every $\epsilon$ we compute the number of samples required to meet the guarantees (using Equations 4 and 5), and independently build 50 subset of the test dataset $\mathcal{D}$. Hence, for each value of $\epsilon$ this give rise to 50 different estimation of global robustness. As $\epsilon$ decreases, the variance of the estimated values decreases as well, quickly converging at around $5\%$ error. The red line in the plot shows the number of samples needed to obtain the required guarantees. Notice that, though the number of samples is exponential wrt $\epsilon$, $5\%$ tolerance is already obtained with just above 700 samples taken from the input distribution. This is orders of magnitudes smaller than test datasets used for training deep networks.

## 4.2 ROBUSTNESS OF DETERMINISTIC NEURAL NETWORKS

In this section, we empirically evaluate the trade-off between global robustness and accuracy for deterministic networks. Recent works have suggested that robustness and accuracy might be at odds, and observed that this was the case when training neural networks on a selection of datasets (Tsipras et al., 2018). While the trade-off between robustness and accuracy is generally different for each application, we contribute to these analyses by quantifying (i.e. with guaranteed statistical error) the robustness/accuracy trade-off in fully connected networks (FCNs) and convolutional neural networks (CNNs) trained on MNIST, Fashion-MNIST (Xiao et al., 2017) and CIFAR-10 using Stochastic Gradient Descent (SGD).

Figure 2 depicts the empirical robustness/accuracy trade-off we observe, computed with statistical error $\epsilon = 0.05$. Overall, the observed trade-off is computed on approximately 3800 different neural network models (each blue dot in the figure represents robustness and accuracy obtained with a particular neural network), namely 1000 fully-connected networks (FCNs) and 1000 convolutional neural networks (CNNs) for MNIST, 500 FCNs and 500 CNNs for Fashion MNIST, and 800 CNNs for CIFAR-10. The different networks were obtained by means of a grid search over the hyperparameters space, where we vary depths, widths, activation functions, learning rates and training epochs. In the case of CNNs we also vary the number of convolutional layers, number of filters in each convolutional layer, and the size of the kernel used (additional details about the hyperparameters used and values explored can be found in the Appendix Section B). For CIFAR-10 approximately 200 models are taken from the DEMOGEN model dataset (Jiang et al., 2018). The global properties checked in these analyses are global robustness induced by: checking whether NNs prediction on $L_\infty$ $\delta$-ball produces confidence variations above $0.50$ (shown in Figures (a) and (c)); computing maximum softmax variation in $L_\infty$ norm (shown in Figures (b), (d) and (e)); checking for changes in classification (shown in Figure (c)). These are further reported in each figure box. Notice that the $\delta$ used for Fashion-MNIST and CIFAR-10 is smaller than the one used for MNIST, as every network was found fully non-robust for larger values of $\delta$. We find that for harder problems, the network performance is more varied which results in greater noise in the quantification of the trade-off (as seen in Figure 2).

For each of the datasets analysed and for all tested measures of robustness, we find a negative correlation between accuracy and robustness. This supports the current hypothesis of the robustness/accuracy trade off, and demonstrates that optimizing hyperparameters for accuracy might have a negative effect on the resulting network robustness. We perform empirical analysis to find the hyper-parameters that have a most profound effect on the network robustness. These are shown in the boxplots in Figure 2, which are computed by averaging out the results across all the networks tested. We find that both increasing model capacity (purple boxplots) and increasing training duration (green boxplots) tend to have a negative effect on robustness. Interestingly, these suggest a relationship between adversarial vulnerability and overfitting. We remark, however, that, though they align with previous hypothesis and similar empirical results (Tsipras et al., 2018), these results are empirical, and inevitably based on observations from only finitely many networks.

## 4.3 EFFECT OF ITERATIVE MAGNITUDE PRUNING ON ROBUSTNESS

Iterative Magnitude Pruning (IMP) is a network compression technique which has shown, empirically, to be effective at reducing the number of weight parameters in neural networks. Empirical analyses demonstrated that IMP can at times remove up to $99\%$ of the weights, while keeping (and even improving on) the original network test accuracy (Frankle & Carbin, 2018). As the existence of

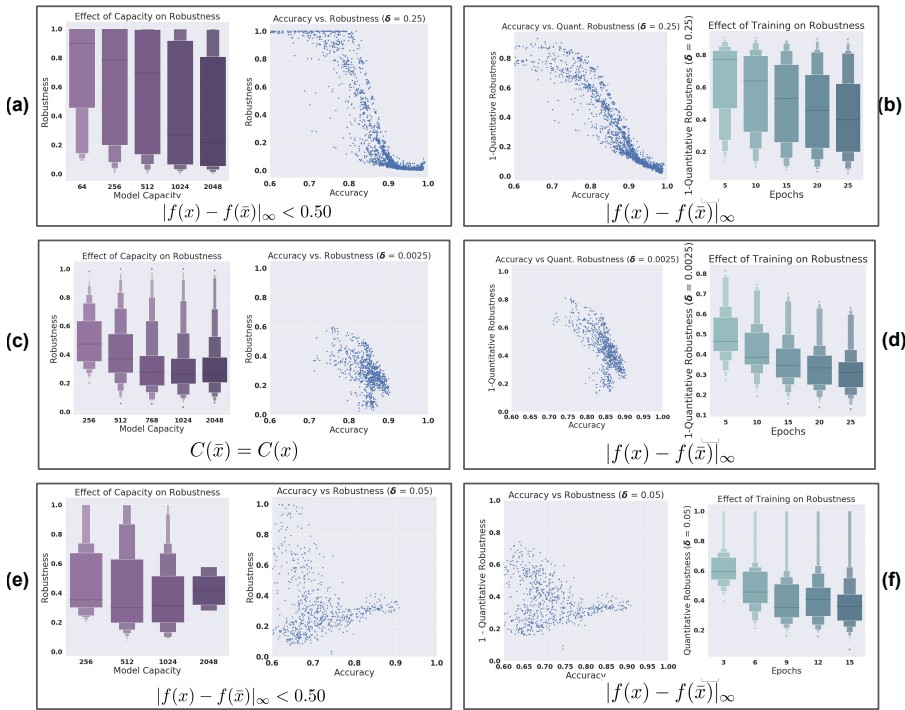

Figure 2: Analysis of empirical accuracy/robustness trade-off computed on 4000 deterministic FCNs and CNNs (each blue dot in the center plots represents the accuracy/robustness result for each network). Each set of plots has been labeled with the functions used to compute robustness. **First row:** analyses on the MNIST dataset. **Second row:** analyses on the Fashion-MNIST dataset. **Third row:** analyses on the CIFAR-10 dataset. We observe a negative trend between accuracy and robustness.

a non-trivial interplay between model capacity and adversarial robustness has been suggested in the literature (Goodfellow et al., 2014), we here explore the effect of IMP on global robustness on the same architectures described in Section 4.2 for MNIST and Fashion-MNIST. Results for this analysis are shown in Figure 3, where we show the effect of IMP when removing $50\%, 75\%, 90\%$ and $99\%$ of the weights (IMP results reported in green dots and normal training results reported using blue dots in (a) and (b)). We find for IMP a similar trade-off to the one observed when pruning is not applied, with a slight reduction in accuracy when severe pruning strength is applied (shown in (c)). As the boxplots show, we find that the amount of pruning has no statistically significant effect on any of the global robustness measures investigated here. These observations are inline with the impact of other notions of weight regularization observed in (Goodfellow et al., 2014), further highlighting a relationship between weight pruning and network regularisation.

## 4.4 ROBUSTNESS OF BAYESIAN NEURAL NETWORKS

In this section we explore the global robustness of networks trained in Bayesian settings, which in principle should not suffer from overfitting-like issues (Gal & Smith, 2018). We perform approximate Bayesian training on MNIST and Fashion-MNIST using Hamiltonian Monte Carlo (HMC), which is the gold-standard for Bayesian inference (Neal et al., 2011). Unfortunately, this does not allow us to scale to CIFAR-10. While this could be possible by using other Bayesian approximate training techniques, it would add a non-trivial interplay between the results observed and the quality of the approximation. We again explore the model architecture space using the same hyper-parameters discussed in Section 4.1 (instead of varying learning rates and epochs, we vary the parameters of the numerical integrator for the Hamiltonian dynamics and the number of samples that we use in order to approximate the posterior distribution). As such, the explored architectures are exactly the same as those explored for deterministic NNs and the only difference lies hence in the training methods (that is, deterministic vs. Bayesian). Results for this analysis are given Figure 4, with pink dots showing Bayesian results and blue dots the deterministic ones (as in Figure

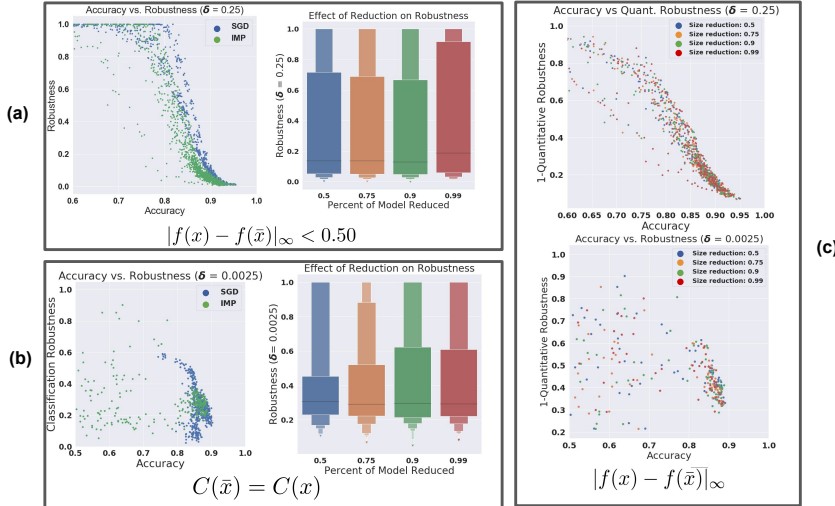

Figure 3: Quantitative analysis of global adversarial robustness on models trained with iterative magnitude pruning. **(a):** robustness results on MNIST; **(b):** robustness results on Fashion-MNIST; **(c):** quantitative robustness on MNIST (top) and Fashion-MNIST (bottom). We observe that pruning leads to comparable trade-off wrt when pruning is not applied. The boxplots show a lack of correlation between the proportion of weights pruned and global robustness.

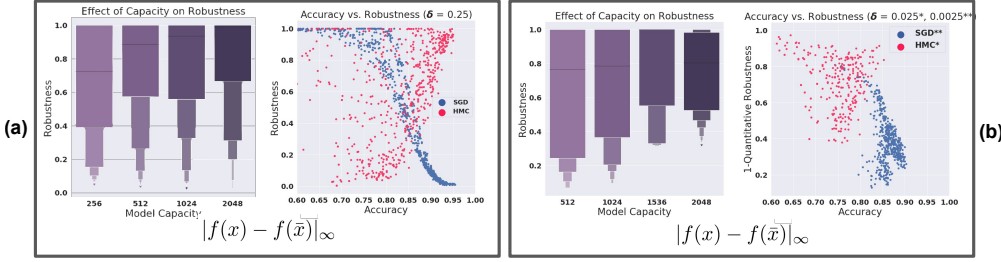

Figure 4: Global robustness analysis of Bayesian NNs. **(a):** results on MNIST; **(b):** results on Fashion-MNIST. Each plot is labeled with the notion of robustness used. In the rightmost plot, SGD was measured with attacks an order of magnitude weaker as the models were more vulnerable.

2). Interestingly, we find that, not only do Bayesian networks not exhibit the negative correlation between robustness and accuracy, they in fact reverse this correlation. This shows that, for the architectures analysed here, and for MNIST and Fashion-MNIST, robustness and accuracy are not at odds. Actually, the results suggest that selecting a Bayesian network hyper-parameters to optimise test accuracy leads to more globally robust networks as well. We further inspect the effect of model capacity on the global adversarial robustness. Interestingly, we show that there is a weak positive correlation between model capacity and robustness for both MNIST and Fashion-MNIST, which again reverses the trend observed for deterministic networks.

## 5 CONCLUSION

We considered a probabilistic measure of global adversarial robustness of neural networks and gave methods for its estimation with a-priori statistical guarantees. The presented techniques were employed to provide statistically sound estimates of the robustness profile of an array of neural network architectures on MNIST, CIFAR-10 and Fashion-MNIST. We further investigated how the robustness/accuracy trade-off is affected by different training approaches, including Bayesian and iterative pruning methods. The methods discussed here rely only on the continuity of neural network models, and thus generalise to any machine learning model characterised by a continuous function, provided that the given notion of local robustness can be computed in practice.

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

## A    PROOFS

In this Section of the Appendix we provide proofs for the Propositions and Theorems stated in the main paper.

**Proof of Theorem 1** By Definition we have

$$Prob(|R^{emp}(\mathbf{g}, S) - R(\mathbf{g})| > \epsilon) =$$
$$Prob(|\sum_{x \in S} \frac{\mathbf{g}(x)}{|S|} - \mathrm{E}_{x \sim P}[\mathbf{g}(x)]| > \epsilon)$$

Since $\mathbf{g}(x)$ is an indicator function, and under the assumption that $\mathbf{g}$ is measurable, $\mathrm{E}_{x \sim P}[\mathbf{g}(x)]$ defines the probability of a random variable taking values in $\{0, 1\}$. Hence, the difference between $\mathrm{E}_{x \sim P}[\mathbf{g}(x)]$ and its empirical frequency $\sum_{(x,y) \in S} \frac{\mathbf{g}(x)}{|S|}$ can be upper bounded by employing additive Chernoff bounds (Vapnik (2013)), yielding Equation 4.

**Proof of Theorem 2**

$$Prob(|R^{emp}(\mathbf{g}, S) - R(\mathbf{g})| > \epsilon)$$
$$= Prob(R^{emp}(\mathbf{g}, S) - R(\mathbf{g}) > \epsilon \vee R^{emp}(\mathbf{g}, S) - R(\mathbf{g}) < -\epsilon)$$
$$= Prob(R^{emp}(\mathbf{g}, S) - R(\mathbf{g}) > \epsilon \vee R(\mathbf{g}) - R^{emp}(\mathbf{g}, S) > \epsilon)$$
$$\text{(By the union bound)}$$
$$\leq Prob(R^{emp}(\mathbf{g}, S) - R(\mathbf{g}) > \epsilon) + Prob(R(\mathbf{g}) - R^{emp}(\mathbf{g}, S) > \epsilon)$$

Each of these terms can now be bounded by using Hoeffding's inequalities (Vapnik, 2013).

**Proof of Proposition 1** For $x \in \mathbb{R}^m$ and $j \in \{1, ..., n\}$, define

$$r_j(x) = \min_{y \in T^x} \left( f_j(y) - \max_{i \neq j} f_i(y) \right).$$

Note that, by definition of $r_j$ and of g, it holds that $\mathbf{g}(x) = 1$ and $C(x) = \{j\}$ if and only if $r_j(x) > 0$. Furthermore, $r_j$ is measurable by the Measurable Maximum Theorem (Theorem 18.19 in (Guide (2006))). It thus follows that the set

$$X_j = \{x \in \mathbb{R}^m \mid r_j(x) > 0\}$$

is measurable, being it a super-level set of a measurable function (hence, the counter-image via a measurable function of a Borel-measurable set), and so is its indicator function $\mathbf{1}_{X_j}(x)$. Measurability of g then follows from the fact that

$$\mathbf{g}(x) = \sum_{j=1}^{n} \mathbf{1}_{X_j}(x).$$

**Proof of Proposition 2**

$l : \mathbb{R}^m \times \mathbb{R}^m \to \mathbb{R}$ is a Carathèodory function. Hence, measurability of $\bar{\mathbf{g}}$ follows from the Measurable Maximum Theorem (Theorem 18.19 in (Guide (2006))).

## B  EXPERIMENTAL SETTINGS

In this Section of the appendix we provide details on the training of the NN models that are discussed in Section 4 of the main paper.

For all networks, save those for CIFAR10, we use standard stochatic gradient descent with the noted learning rates, for CIFAR10, in order to speed up convergence we use the Adam optimizer. For HMC, we approximate the posterior using built-in functions provided by Tensorflow and Edward.

For lottery ticket, we implemented a general iterative magnitude pruning method which works with models written in Keras. Source code for this method is supplied in the project repository.

Below, we list the hyperparameter grid searchs that was performed in order to generate the plots found in the paper.

### B.1  MNIST - SGD - FCNs

| Parameter | Values |
|---|---|
| Depth | 1, 2, 3 |
| Width | 64, 128, 256, 512 |
| Activation Function | ReLu, Tanh, Sigmoid |
| Learning Rates | 0.05, 0.01, 0.005, 0.001 |
| Epochs | 5, 10, 15, 20, 25 |

### B.2  MNIST - SGD - CNNs

| Parameter | Values |
|---|---|
| FC Depth | 1, 2 |
| FC Width | 128, 256 |
| Activation Function | ReLu |
| Convolutional Filters | 5, 10, 20, 30 |
| Convolutional Kernel Size | 3, 5, 7 |
| Convolutional Depth | 1, 2, 3 |
| Learning Rates | 0.05, 0.01, 0.005, 0.001 |
| Epochs | 3, 6, 9, 12, 15 |

### B.3 FASHION MNIST - SGD - FCNS

| Parameter | Values |
|---|---|
| Depth | 2, 3, 4 |
| Width | 256, 512, 1024 |
| Activation Function | ReLu, Tanh, Sigmoid |
| Learning Rates | 0.01, 0.005, 0.001 |
| Epochs | 5, 10, 15, 20, 25 |

### B.4 FASHION MNIST - SGD - CNNS

| Parameter | Values |
|---|---|
| FC Depth | 1, 2 |
| FC Width | 128, 256 |
| Activation Function | ReLu |
| Convolutional Filters | 5, 20, 30 |
| Convolutional Kernel Size | 3, 5, 7 |
| Convolutional Depth | 1, 2, 3, 4 |
| Learning Rates | 0.0001, 0.001, 0.005 |
| Epochs | 3, 6, 9, 12, 15 |

### B.5 CIFAR-10 - SGD - CNNS

Please refer to the Google DEMOGEN paper in order to extract their parameters for the network-in-network models. Below, where we list convolutional depth we are reporting the number of VGG-style blocks (two consecutive convolutional layers followed by a max pooling layer).

| Parameter | Values |
|---|---|
| FC Depth | 1, 2, 3 |
| FC Width | 128, 256, 512 |
| Activation Function | ReLu |
| Convolutional Filters | 24, 48, 64 |
| Convolutional Kernel Size | 3, 5, 7 |
| Convolutional Depth | 1, 2, 3 |
| Learning Rates | 0.0001, 0.0005, 0.001, 0.005 |
| Epochs | 3, 6, 9, 12, 15 |

### B.6 MNIST - IMP - FCNS

| Parameter | Values |
|---|---|
| Depth | 1, 2, 3 |
| Width | 128, 256, 512, 1024 |
| Activation Function | ReLu, Tanh, Sigmoid |
| Prop. Weight Reduction | 0.5, 0.75, 0.9, 0.99 |
| Iterations of Pruning | 1, 2, 5 |
| Learning Rates | 0.001 |
| Epochs | 5, 10, 15 |

### B.7    FASHION MNIST - IMP - FCNS

| Parameter | Values |
|---|---|
| Depth | 1, 2, 3, 4 |
| Width | 256, 512 |
| Activation Function | ReLu, Tanh, Sigmoid |
| Prop. Weight Reduction | 0.5, 0.75, 0.9, 0.99 |
| Iterations of Pruning | 1, 2, 5 |
| Learning Rates | 0.001 |
| Epochs | 5, 10, 15 |

### B.8    MNIST - HMC - FCNS

| Parameter | Values |
|---|---|
| Depth | 1, 2, 3 |
| Width | 128, 256, 512 |
| Activation Function | ReLu, Tanh, Sigmoid |
| Step Size | 0.01, 0.005, 0.001, 0.0001 |
| Number of Steps | 10, 15, 20 |
| Samples from Posterior | 100, 200, 400, 600 |

### B.9    FASHION MNIST - HMC - FCNS

| Parameter | Values |
|---|---|
| Depth | 1, 2, 3 |
| Width | 128, 256, 512 |
| Activation Function | ReLu |
| Step Size | 0.01, 0.005, 0.001, 0.0001 |
| Number of Steps | 10, 15, 20 |
| Samples from Posterior | 100, 200, 400, 600 |

