# OpenReview forum: "Global Adversarial Robustness Guarantees for Neural Networks"
_ICLR.cc/2020/Conference — Reject_

### Official Review · AnonReviewer1 · 2019-10-22
**Official Blind Review #1**

**Rating:** 1

**Review:**

This paper seeks to analyze the global robustness of neural networks, a concept defined in the paper. The authors show using concentration inequalities that the empirical local robustness approximates the global robustness. The authors investigate various other issues in the robustness literature, including the robustness/accuracy tradeoff, whether iterative pruning increases robustness, and the robustness of Bayesian networks.

I would vote for rejecting this paper for two key reasons. First, the notion of global robustness is not well-motivated (why do we want to compute this metric? What does it tell us that local robustness does not?). Second, the paper tries to do too many different things, and as a result does not give enough attention to any particular topic.

First, I believe it is up to the authors to motivate their study of global robustness further. While I acknowledge that a few prior works exists along these lines, I do not feel that this work provides much new insight into why global robustness is interesting to examine.

The authors go on to prove results showing that an empirical estimator of the local robustness will converge to the global robustness. The bounds require that the dataset size scales with eps^-2, where eps is the error. This is not terrible but also not great; for example, achieving 1% error requires a dataset size of 10^4 (realistically, even larger datasets would be required to achieve results with high probability).

Next, I would suggest that the authors avoid using the word “guarantees” if they are estimating empirical local robustness in an approximate (rather than exact) manner. Guarantees implies strict results, but the authors use a weak attack (FGSM) to approximate empirical local robustness. The results from FGSM could be far from optimal; the authors could use a stronger attack (e.g. PGD) in addition to changing the wording, or they could find provable guarantees using alternate methods.

Lastly, the authors try to tackle 3 extra questions beyond global robustness toward the end of the paper, and the last two questions are not properly fleshed out.

I like section 4.2, where the authors empirically show that networks that have better hyperparameters (for regular accuracy) tend to be less robust. This is a confirmation of a previously studied phenomena in the literature. Ideally, I would also appreciate it if the authors found the line of best fit to the dataset in addition to the plots provided. I would like a clarification on whether any of these networks were trained to be robust, although it appears that they were all trained normally. I would also like to see plot 2c (for the standard case of robustness of C(x_tilde) = C(x)), except for MNIST and CIFAR10 as well. I feel that the last-layer representation metric the authors analyze (f(x) is close to f(x_tilde)) could be misleading, as robustness on the last layer does not necessarily imply standard adversarial robustness.

Section 4.3 explores iterative pruning, but that seems fairly unrelated to the rest of the paper. Finally, Section 4.4 tries to show the opposite trend for Bayesian Neural Networks, but unfortunately the results for such networks do not yet scale beyond MNIST.

Additional Feedback:

- Why did you use R^emp and D^emp as opposed to just R^emp(g) and R^emp(g_bar)?
- In Figure 1, what is the dataset size |S|?
- In the last sentence of Section 4.3, I didn’t understand what you meant about the relationship between weight pruning and network regularization. Do you mean that weight regularization has no effect on robustness, just like iterative weight pruning?


**Experience Assessment:**

I have published one or two papers in this area.

**Review Assessment: Checking Correctness Of Derivations And Theory:**

I did not assess the derivations or theory.

**Review Assessment: Checking Correctness Of Experiments:**

I assessed the sensibility of the experiments.

**Review Assessment: Thoroughness In Paper Reading:**

I read the paper at least twice and used my best judgement in assessing the paper.

---

> ### Author Response · Authors · 2019-11-13
> **Responses to Reviewer #1 (Part 1)**
>
> Reviewer:  First, the notion of global robustness is not well-motivated (why do we want to compute this metric?  What does it tell us that local robustness does not?).  While I acknowledge that a few prior works exists along these lines, I do not feel that this work provides much new insight into why global robustness is interesting to examine.
>
> Response:
> 	Local robustness is a property specific to a given test point, and thus, in general, does not give any information about the robustness of the NN in other test points. On the other hand, global robustness is the probability that a point sampled from the data distribution is robust to adversarial perturbation  (i.e.,  it  is  locally  robust), and  thus  a  property  of  the  network  and  of  the  particular problem only.  Global robustness is therefore useful as a model selection criterion, enabling the choice of a model that is not only accurate, but also robust on unseen inputs.  As models are compared in terms of accuracy, global robustness enables us to compare models in terms of their robustness wrt adversarial examples.    Building on this, in this paper we develop a framework for the computation of global robustness up to any a-priori statistical error, and employ it to compare different neural network architectures and training paradigms in terms of their robustness.
>
> Reviewer:  the paper tries to do too many different things, and as a result does not give enough attention to any particular topic.  ...  The authors try to tackle 3 extra questions beyond global robustness toward the end of the paper,  and the last two questions are not properly fleshed out...  Section 4.3 explores iterative pruning, but that seems fairly unrelated to the rest of the paper.  Finally, Section 4.4 tries to show the opposite trend for Bayesian Neural Networks,  but unfortunately the results for such networks do not
> yet scale beyond MNIST
>
> Response:
> 	We would like to stress that the main contribution of this paper lies in the development of a framework for the computation of global adversarial robustness with a-priori statistical guarantees. We first show that these error bounds can be used for neural networks, and then we apply them to investigate the robustness of different neural network architectures and training paradigms.  It is in this way that the investigation into iterative magnitude pruning, which had not been done before, is
> related.  We are empirically studying the effect of network compression on robustness in a way that puts bounds on our error.  Given that this is a comparison of networks where only one variable has changed (the training method), the method proposed in the paper allows us to precisely quantify the expected change in robustness from SGD to IMP.
> 	The decision to consider BNNs was driven by the fact that many of the symptoms of adversarial examples  appear  to  be  from  overfitting  to  the  training  distribution,  and  in  principle  BNNs  do  not suffer from overfitting,  which may result in different trends.  Further,  whereas it is true that BNNs trained  with  HMC  do  not  scale  to  large  datasets  beyond  MNIST,  other  scalable  but  approximate Bayesian  training  methods  exist  (i.e.,  mean-field  variational  inference  and  Monte  Carlo  dropout). We decided to not include these methods in our analysis because they would introduce a non-trivial approximation  error.   However,  we  believe  that  our  analysis  may  lead  to  novel  insights  about  the existence of adversarial examples and how these may be inherently related to training with SGD

---

> ### Author Response · Authors · 2019-11-13
> **Responses to Reviewer #1 (Part 2)**
>
> Reviewer: The bounds require that the dataset size scales with $eps^-2$, where $eps$ is the error. This is not terrible but also not great; for example, achieving 1\% error requires a dataset size of $10^4$ (realistically, even larger datasets would be required to achieve results with high probability).
>
> Response:
> 	We  would  like  to  stress  that,  under  standard  iid  assumptions,  our  guarantees  (i.e., statistical guarantees on the global evaluation of robustness) are formal.  Further, for larger datasets, such as ImageNet with 200,000 held-out images, error less than 1% is easily achievable.  Hence, the statistical error obtained in practice will be always smaller than or equal to the theoretical error given by the bounds in Theorem 1 and 2.  Furthermore, while it is true that our bounds are exponential in $\epsilon^2$ they can still be quite accurate in practice.  This is illustrated empirically in Figure 1 (by using formal local robustness techniques, i.e., guarantees on worst-case behaviour) and discussed in Section 4.1 in the main text.  Moreover, the tightness of Chernoff bounds is discussed in Chapter 4.1 in [Vapnik, V. N. (1998).  Statistical Learning Theory.  Wiley-Interscience.], where it is shown that in case of Bernoulli random variables with probability of a half these bounds cannot be improved.  A way to improve the bounds in Theorem 1 and 2 would be to use sequential sampling schemes [Massart, Pascal.  The tight constant in the Dvoretzky-Kiefer-Wolfowitz inequality.  The Annals of Probability (1990):  1269-1283.]. However, these methods would require substantially fewer samples only when global robustness is close to 0 or 1.  Further, since they are sequential, they are much less straightforward to parallelize (though requiring fewer samples, they may take more time to run in multi-core machines).  Hence, we did not consider these approaches in our paper.
>
>
> Reviewer:  I would suggest that the authors avoid using the word “guarantees” if they are estimating empirical local robustness in an approximate (rather than exact) manner. Guarantees implies strict results, but the authors use a weak attack (FGSM) to approxi- mate empirical local robustness.  The results from FGSM could be far from optimal; the authors could use a stronger attack (e.g.  PGD) in addition to changing the wording, or they could find provable guarantees using alternate methods.
>
> Response:
> 	The bounds of Theorems 1 and 2 provide strict, statistical guarantees on the estimation of any measurable global robustness property.
> That is, independently of the local robustness measure used (e.g., that resulting from FGSM attacks, PGD attacks, or from provable guarantees obtained via formal verification), Theorems 1 and 2 provide formal bounds on the global robustness statistical error specific to that particular local notion.
> Following the reviewer comment, we will modify the paper to stress that the guarantees we provide are for the global estimator, and not for the local robustness property.
> 	Though we agree with the reviewer that formal verification methods for local robustness, which provide provable guarantees, are ideally suited to the task, and indeed we apply them for the experiments reported in  Figure 1 with the MNIST datasets, the computational burden of state-of-the-art verification methods does not allow us to scale to large datasets or to consider analysis on multiple neural network architectures in any reasonable amount of time (e.g., in Figure 2 we evaluated local robustness millions of times, which would be infeasible with formal verification methods).  Similarly to  [Osbert Bastani, et al. Measuring neural net robustness with constraints. NIPS 2016.], we thus compare models in terms of resistance to specific adversarial attacks. We focused on FGSM for scalability, but any other method can be used.
> 	Finally, it is interesting to note that while FGSM (or PGD or any other heuristic attack) does not tell us the exact robustness value for formal local robustness, it does provide us with an upper bound for it. That is, if FGSM finds a small attack that is successful, then the true worst-case attack must be at least that small, if not smaller. Moreover, more sophisticated attacks (e.g. PGD and CW attacks) have been empirically shown to be close to the true worst-case attack [Carlini et. al., Provably Minimally-Distorted Adversarial Examples, 2018]

---

> ### Author Response · Authors · 2019-11-13
> **Responses to Reviewer #1 (Part 3)**
>
> Reviewer:  I would like a clarification on whether any of these networks were trained to be robust, although it appears that they were all trained normally.
>
> Response:
> 	Yes, all of the networks in the paper were trained normally, that is, without adversarial training or robust optimization.  We thank the reviewer for the suggestion, and additional experiments have  been  run.   We  trained  a  set  of  networks  with  robust  optimization  with  an  adversarial  budget of  epsilon  in  the  l-infinity  norm  setting.   We  again  use  FGSM  for  robust  optimization  and  for  the evaluation  (in  order  to  be  consistent  with  the  previous  study  of  SGD).  We  found  that,  for  attacks with a budget less than or equal to epsilon (on the test set) the networks were indeed more robust than their counterparts trained with SGD; however, for attacks with strength greater than epsilon, the robustness-accuracy trade-off still holds.  These networks can thus be contrasted with BNNs, which appear to be more robust as they become more accurate, rather than being trained to be epsilon-robust.
>
>
> Reviewer: I would also like to see plot 2c (for the standard case of robustness of $C(x_tilde) = C(x))$, except for MNIST and CIFAR10 as well. I feel that the last-layer representation metric the authors analyze (f(x) is close to $f(x_tilde))$ could be misleading, as robustness on the last layer does not necessarily imply standard adversarial robustness.
>
> Response:
> 	We  agree  with  the  reviewer  that  the  robustness  notion  over  the  softmax  difference does not directly imply classification robustness.  For DNNs on MNIST we observe that classification robustness is positively correlated until networks reach 85% accuracy, at which point they drop starkly in robustness (from a max classification robustness of 0.40 for networks between 0-85% accuracy to a max classification robustness of 0.17 for networks with greater than 95% classification accuracy).
> 	For CIFAR10,  classification robustness yielded neither positive nor negative correlation with in- crease in accuracy.  On the other hand,  the measure of quantitative robustness yielded a change in robustness that tended to be negatively correlated with accuracy.  We believe that part of the reason for  not  finding  a  strong  correlation  in  this  case  is  to  do  with  the  fact  that  a  majority  of  the  networks achieved less than 80% accuracy, which is before we empirically observe the drop off in previous experiments.
> 	We agree that we should discuss these trends; however, we found the quantitative robustness to be a more rich source of robustness information.  In any case, we will add the suggested plots in the final version of the paper
>
> Reviewer:  In the last sentence of Section 4.3, I didn’t understand what you meant about the relationship between weight pruning and network regularization.  Do you mean that weight regularization has no effect on robustness, just like iterative weight pruning?
>
> Response:
> 	Yes,  that  is  what  we  meant.   It  has  been  shown  in  several  previous  works,  including [Goodfellow et al.  Explaining and Harnessing Adversarial Examples.], that standard regularization, which  focuses  on  magnitude  reduction  of  weights,  was  shown  empirically  to  have  no  effect  on  the adversarial robustness of models.  This is precisely what we found for iterative weight pruning.  So, despite this method regularizing the network size in a different way, we nonetheless observe the same phenomenon.

---

### Official Review · AnonReviewer2 · 2019-10-23
**Official Blind Review #2**

**Rating:** 1

**Review:**

Summary:

The paper formally defines local and global adversarial robustness. Following that, the paper investigates how to estimate local and global adversarial robustness using an estimator based on evaluating these quantities on the empirical distribution. Using a Chernoff bound, the papers evaluate probabilistic bounds on the deviation of the estimated quantities from the true quantities.  Finally, simulations are provided to evaluate these bounds for examples.

Comments:

The authors' insistence on their contribution being proving measurability does not make sense -- of course everything is measurable! Furthermore, the formal definitions or local and global robustness are well-known, the bounds in Theorems 1 and 2 are not novel and highly unlikely to be tight. The redeeming aspect of the paper is the experiments, where the authors show that these bounds can actually be (approximately) calculated. However, I feel that merely experimental results with correct but not significant theoretical contributions does not meet the bar for acceptance.

**Experience Assessment:**

I have read many papers in this area.

**Review Assessment: Checking Correctness Of Derivations And Theory:**

I assessed the sensibility of the derivations and theory.

**Review Assessment: Checking Correctness Of Experiments:**

I assessed the sensibility of the experiments.

**Review Assessment: Thoroughness In Paper Reading:**

I made a quick assessment of this paper.

---

> ### Author Response · Authors · 2019-11-13
> **Responses to Reviewer #2**
>
> Reviewer: The authors’ insistence on their contribution being proving measurability does not make sense – of course everything is measurable!
>
> Response:
> 	We would like to disagree on this point.  While it is true that the main contributions of the paper do not lie in the measurability proof, and that  measurability of the global robustness property for neural networks is expected, this has to be shown explicitly.  It is not possible to substantiate the claim on statistical guarantees by any other means. In particular, we have to formally establish the measurability of $g$ and $\bar{g}$ to ensure that we are dealing with well defined random variables. Without this, we cannot guarantee the validity of the bounds in Theorem  1 and 2. See for instance Chapter 5 in [Richard M. Dudley. Uniform central limit theorems. No. 63. Cambridge university press, 1999].
>
>
> Reviewer: The  formal  definitions  of  local  and  global  robustness  are  well-known,  the bounds in Theorems 1 and 2 are not novel and highly unlikely to be tight.
>
> Response:
> 	We agree with the reviewer that definitions of local and global robustness are already known and used in practice.  This is the reason why in this paper we propose a framework to study these quantities with statistical guarantees.  In fact, although concentration inequalities and Chernoff bounds are well known techniques, these have been not been used before to establish global robustness. Moreover, Theorems 1 and 2 can be quite accurate in practice and the resulting number of samples required is generally under control.  This is illustrated empirically in Figure 1 and discussed in Section 4.1  in  the  main  text.   Moreover,  the  tightness  of  Chernoff  bounds  is  discussed  in  Chapter  4.1  in [Vapnik,  V. N. (1998).  Statistical Learning Theory.  Wiley-Interscience.],  where it is shown that in case of Bernoulli random variables with probability of a half these bounds cannot be improved.
>
>
> Reviewer:  The  redeeming  aspect  of  the  paper  is  the  experiments,  where  the  authors show that these bounds can actually be (approximately) calculated.  However, I feel that merely  experimental  results  with  correct  but  not  significant  theoretical  contributions does not meet the bar for acceptance.
>
> Response:
> 	We would like to stress that the main contribution of this paper lies in the development of a framework for the computation of global adversarial robustness with a-priori statistical guarantees.  We first show that these bounds can be used for neural networks, and then we apply them to investigate the  robustness  of  different  neural  network  architectures  and  training  paradigms. This  allows  us  to confirm and quantify previously reported results on the trade-off between generalisation accuracy and adversarial robustness of neural networks. We then investigate the relationship between model capacity and model robustness in iterative magnitude pruning settings.  We find that weight pruning does not increase  robustness  despite  greatly  reducing  model  capacity.   Finally,  we  evaluate  the  robustness  of Bayesian neural networks and compare it with their deterministic counterpart, which, to the best of our knowledge, had never been evaluated.  Our finding that BNNs are more robust wrt gradient based attacks,  in  our  opinion,  warrants  further  exploration  into  the  use  of  BNN  models  in  safety-critical scenarios.

---

### Official Review · AnonReviewer3 · 2019-10-24
**Official Blind Review #3**

**Rating:** 3

**Review:**

This paper studies the adversarial robustness of neural networks by giving theoretical guarantees, providing statistical estimators and running experiments. It is a lot of work and it is reasonably written. The problem is that a fair bit of it is quite basic: for example the measurability property is very much expected -- noone was doubting it, and the proof is more of a formality than a contribution. Similarly with the statistical sampling: the method seems to rely on i.i.d. sampling -- has this reviewer missed any important details? If not, then it's only the bounds that are a contribution, but the method is not. We would appreciate more specific description of the main contribution, without it we cannot recommend the acceptance of this paper.

I am very grateful to the authors for their response. I feel now that a main weakness of this paper may be that it puts too many results in one place. I would strongly suggest re-writing it, possibly into separate papers, to make the things pointed out in the response more clear and self-standing.

**Experience Assessment:**

I have read many papers in this area.

**Review Assessment: Checking Correctness Of Derivations And Theory:**

I assessed the sensibility of the derivations and theory.

**Review Assessment: Checking Correctness Of Experiments:**

I carefully checked the experiments.

**Review Assessment: Thoroughness In Paper Reading:**

I read the paper at least twice and used my best judgement in assessing the paper.

---

> ### Author Response · Authors · 2019-11-13
> **Responses to Reviewer #3**
>
> Reviewer:  the measurability property is very much expected – no one was doubting it, and the proof is more of a formality than a contribution.
>
> Response:
> 	We agree with the reviewer that measurability of global robustness for neural networks can be expected.
> Nevertheless, we argue that this has to be shown explicitly. In particular, we have to formally establish the measurably of $g$ and $\bar{g}$ to guarantee that we are dealing with well defined random variables. Without this, we cannot guarantee the validity of the bounds in Theorem  1 and 2. For further details on this issue, see  Chapter 5 in [Richard M. Dudley. Uniform central limit theorems. No. 63. Cambridge university press, 1999].
>
> Reviewer:  has  this  reviewer  missed  any  important  details?   If  not,  then  it’s  only  the bounds that are a contribution, but the method is not. We would appreciate more specific description of the main contribution, without it we cannot recommend the acceptance of this paper.
>
> Response:
> 	While Chernoff bounds are well known, the main contribution of this paper lies in the development of a framework for the computation of global adversarial robustness with a-priori statistical guarantees (provided by the Chernoff's bound), which we use to investigate the robustness of different neural network architectures and training paradigms. This allow us to confirm and quantify previously reported results on the trade-off between generalisation accuracy and adversarial robustness of neural networks, demonstrated through a large-scale study. We then investigate the relationship between model capacity and model robustness in iterative magnitude pruning settings. We find that weight pruning does not increase robustness despite greatly reducing model capacity. Finally, we evaluate the robustness of Bayesian neural networks and compare it with their deterministic counterparts, which, to the best of our knowledge, had never been evaluated. Our finding that BNNs are more robust wrt gradient-based attacks, in our opinion, warrants further exploration into the use of BNN models in safety-critical scenarios.

---

### Decision · Program_Chairs · 2019-12-19

**Decision:**

Reject

**Comment:**

The authors propose a framework for estimating "global robustness" of a neural network, defined as the expected value of "local robustness" (robustness to small perturbations) over the data distribution. The authors prove the the local robustness metric is measurable and that under this condition, derive a statistically efficient estimator. The authors use gradient based attacks to approximate local robustness in practice and report extensive experimental results across several datasets.

While the paper does make some interesting contributions, the reviewers were concerned about the following issues:
1) The measurability result, while technically important, is not surprising and does not add much insight algorithmically or statistically into the problem at hand. Outside of this, the paper does not make any significant technical contributions.
2) The paper is poorly organized and does not clearly articulate the main contributions and significance of these relative to prior work.
3) The fact that the local robustness metric is approximated via gradient based attacks makes the final results void of any guarantees, since there are no guarantees that gradient based attacks compute the worst case adversarial perturbation. This calls into question the main contribution claim of the paper on computing global robustness guarantees.

While some of the technical aspects of the reveiwers' concerns were clarified during the discussion phase, this was not sufficient to address the fundamental issues raised above.

Hence, I recommend rejection.